# Biliary Atresia in 2021: Epidemiology, Screening and Public Policy

**DOI:** 10.3390/jcm11040999

**Published:** 2022-02-14

**Authors:** Richard A. Schreiber, Sanjiv Harpavat, Jan B. F. Hulscher, Barbara E. Wildhaber

**Affiliations:** 1Division of Gastroenterology, Hepatology and Nutrition, Department of Pediatrics, Faculty of Medicine, University of British Columbia, Vancouver, BC V6T 1Z3, Canada; 2Division of Gastroenterology, Hepatology and Nutrition, Department of Pediatrics, Baylor College of Medicine and Texas Children’s Hospital, Houston, TX 77030, USA; harpavat@bcm.edu; 3Department of Surgery, Division of Pediatric Surgery, University Medical Center Groningen, University of Groningen, 9713 GZ Groningen, The Netherlands; j.b.f.hulscher@umcg.nl; 4Swiss Pediatric Liver Center, Division of Pediatric Surgery, Department of Pediatrics, Gynecology, and Obstetrics, University of Geneva, 1205 Geneva, Switzerland; barbara.wildhaber@hcuge.ch

**Keywords:** biliary atresia, pediatric liver disease, newborn screening, public health, epidemiology

## Abstract

Biliary atresia (BA) is a rare newborn liver disease with significant morbidity and mortality, especially if not recognized and treated early in life. It is the most common cause of liver-related death in children and the leading indication for liver transplantation in the pediatric population. Timely intervention with a Kasai portoenterostomy (KPE) can significantly improve prognosis. Delayed disease recognition, late patient referral, and untimely surgery remains a worldwide problem. This article will focus on biliary atresia from a global public health perspective, including disease epidemiology, current national screening programs, and their impact on outcome, as well as new and novel BA screening initiatives. Policy challenges for the implementation of BA screening programs will also be discussed, highlighting examples from the North American, European, and Asian experience.

## 1. Introduction

Biliary atresia (BA) is a rare orphan newborn liver disease that results from an idiopathic progressive fibrosclerosing obliteration of large bile ducts [1,2]. The condition is recognized as one of the most rapidly progressive liver diseases known to man. It is the leading cause of liver-related death in children and the foremost indication for liver transplantation in the pediatric population. BA clinically manifests in the first few weeks of life with jaundice and acholic pale stools, the prototypical clinical features of an obstructive-type jaundice, associated with the biochemical hallmark of serum conjugated (direct) hyperbilirubinemia. The current standard of care for BA is sequential surgery with an initial Kasai hepato-portoenterostomy (KPE), in which the obstructed bile duct is resected and a loop of the small bowel is brought to the porta hepatis of the liver to restore bile flow, followed by liver transplantation for those in whom the KP fails or who progress to cirrhosis and liver failure at a later pediatric age or into adulthood. Without any surgical intervention, all infants with BA will die by three years of age. The aims of this article are to provide up-to-date knowledge of the epidemiology of biliary atresia, to focus on the current clinically applicable BA screening strategies implemented in large populations or national programs, and to describe the challenges of BA screening in the context of public health policies.

## 2. Epidemiology and Pathogenesis

BA is a worldwide disease affecting multiple ethnicities. In a recent comprehensive review, the incidence of BA was shown to range widely among countries reporting population-based data, from approximately 1:5000 newborns in Taiwan to 1:20,000 in Europe, Canada, and areas in the USA [3,4]. The highest rates (1:3500) were reported from French Polynesia. Data from several Western countries and regions in the USA or from developing countries is sparse—if not absent—as depicted in Figure 1.

BA is now recognized to have several clinical phenotypes, including isolated BA, syndromic BA with other malformations, cystic-type BA, and BA in association with the cytomegalovirus (CMV). Each of these may have differing etiologic and pathophysiologic mechanisms which could influence the regional frequency of the disease. For example, in Asia, only 2% of the BA cases have the syndromic phenotype, whereas in the West, up to 20% of BA cases are syndromic [4]. A recent Chinese report identified as many as 50% of BA cases being CMV-positive, whereas the UK found it was 10% in their series [5,6].

A “two-hit” theory has long been hypothesized for the pathogenesis of the isolated BA phenotype, with an initial viral infection followed by an exaggerated immune response inducing sclerosing cholangiopathy [7,8]. Several viruses have been shown to be implicated, including the rotavirus, reovirus, and CMV, but none has been conclusively linked to BA. Recently, the isoflavonoid “biliatresone” from the *Dysphania* plant has been invoked as a putative environmental factor in pathogenic disease models [9].

### 2.1. Seasonal Variability

If the origin of the isolated BA phenotype is attributable to an environmental/infectious insult during pregnancy or the perinatal period, one might expect case-clustering in time and space. In contrast, those with a syndromic BA phenotype might be more heavily influenced by genetic determinants. Reports on the epidemiology of BA are variable. In the recent comprehensive literature review by Jimenez–Rivera, 11 out of the 40 papers studied (27.5%) investigated seasonal variations in incidence [3]. Only two studies found significant seasonal variations: an increased incidence of BA from August to October in a southern US city and December to March in a southeastern US city, while the other papers did not find such seasonal variation, or at least, not any statistically significant clustering [10,11]. A recent paper from Korea which was not incorporated in the review supported clustering in summer, while on the other hand, recent results from the Netherlands, incorporating all BA cases in the country in the last three decades, did not observe any temporal clustering for isolated BA [12,13]. These contrasting results may be attributed to geographic or ethnic diversity, varying BA phenotypes, different pathogens, or other yet unknown factors. Interestingly, a weak yet statistically significant correlation between the incidence of maternal infection at the time of conception, including *Chlamydia trachomatis*, and the subsequent development of BA was observed in a recent Dutch study [13].

### 2.2. Geographic Variations

There clearly are worldwide differences in BA incidence. However, there might also be differences within one country or region. For instance, among the English and Welsh cohorts, geographical variations have been found along a northwest/southeast axis varying from 0.38 (northwest England) to 0.78 (southeast England)/10,000 live births [14]. In the Netherlands, there was a 68% increased incidence for isolated BA in rural areas when compared to urban areas, a difference not seen with syndromic BA [13]. This is in line with previous reports from Texas and Sweden, but in contrast to findings reported in the New York state [10,15,16]. Interestingly, several centers have observed a lower incidence of BA cases during the COVID-19 pandemic, when strict public health guidelines for social distancing, mask-wearing, and frequent handwashing were in place [17]. A study to evaluate BA incidence during the COVID-19 pandemic is currently underway in Europe (European Reference Network RARE LIVER). In conclusion, there is contradicting evidence regarding seasonal, as well as geographic clustering of BA, even within one area. Genetic and environmental factors may account for these observed epidemiologic variations, but these determinants are not well-defined. Large-scale international studies are needed to answer these epidemiological conundrums.

## 3. Early Intervention Is a Key Prognostic Indicator: The Need for Screening

The initial KPE operation, first reported in 1959 in Japan by Dr. Morio Kasai, was adopted as a realistic life-saving operation for affected infants in the Western world in the 1970s [18,19]. Further experience recognized that the success of the KPE, defined as the clearance of jaundice and normalization of the serum bilirubin by six months after surgery, correlated best with infant age at the time of surgery [20,21]. A successful KPE postpones the need for early urgent liver transplantation in infancy and gives the potential for longer native liver survival well into adulthood. For example, with surgery before 60 days of age, over 70% of patients become jaundice-free, and 75% of these cases have 10-year survival rates with their native liver [22,23,24,25]. In contrast, late KPE intervention after 90 days of age has a worse prognosis, with fewer than 25% of cases having 4- to 5-year native liver survival [26,27]. In these late-presenting cases, many centres instead defer the initial KPE and proceed directly to liver transplantation. However, reports from the UK and France gave evidence that the KPE could be successfully performed in infants older than 3 months of age, obviating the need for early liver transplantation, provided there are no preoperative signs of hepatic decompensation or severe portal hypertension [28,29].

### 3.1. KPE at Infant Age < 30 Days Is Optimal

Historically, several US and European experts advocated that a KPE performed at too early an age (<30 days) was ineffective [30]. The prevailing view suggested the optimal timing for the KPE was at 45–60 days of age. This ‘old-school’ dogma—that an age at KPE which is too young portends a much worse prognosis—has been conclusively disproven and should be abandoned. Recent studies have firmly demonstrated that KPE intervention at <30 days of age achieves best results and prolongs the native liver survival even into adulthood [26,27]. The largest series reported, including 1428 patients, showed that the 25-year survival rate with native liver was 38%, 27%, 22%, and 19% in patients who had their KPE in the first, second, and third months of life or later, respectively [25]. Japanese, Canadian, and French national studies have shown that optimal rates for native liver survival post-KPE were at a very young infant age, with the best outcomes in those who received the KPE at <30 days of age [22,26,27]. Importantly, an early KPE does not pose additional risks and is not associated with more complications compared with a KPE in older babies. Infant age and weight at the time of the operation are not significantly correlated with adverse events [31].

### 3.2. The Problem of ‘Late’ Referral

While all evidence points towards an operative KPE strategy of ‘the sooner the better’, children with BA often come to the attention of a pediatric gastroenterologist and/or a pediatric surgeon at a ‘late’ age. Over the last decade, the median age at KPE in Western countries has been at around 60 days, which implies that some 50% of cases do not have the KPE before that age and fail to meet the international quality criterium of <60 days of age [22,24,25,26,32]. The age of KPE has not improved over the last several decades [33]. The average age at KPE may be even older in some regions of Asia, South Asia, Africa, and South America, although comprehensive data are lacking.

There are several major obstacles to early disease recognition. Jaundice in newborns is considered a benign process most often associated with “breast milk jaundice”, and further investigations are not pursued by newborn health care providers. This practice persists despite guideline recommendations by global expert panels and pediatric societies worldwide to test serum total and direct or conjugated bilirubin in all infants with persistent jaundice for more than two to three weeks. The monitoring for pale stools by health care providers or parents is not routine to standard well-baby care. Additionally, in several jurisdictions, the schedule for routine well-baby visits (within two weeks after birth and then at the first vaccination at two months of age) misses the ‘window of opportunity’ for early case identification.

## 4. Newborn Screening for Biliary Atresia

One solution to address the problem of late referral for BA is newborn screening. BA satisfies disease-specific criteria for newborn screening, as elaborated by the WHO guidelines. The condition is an important public health problem for infants, families, and the community at large. Without early detection and intervention, there is a likelihood for significant morbidity and mortality at a young age, with the potential need for urgent liver transplantation in infancy. This trajectory has a significant impact on the child and family, as well as caregivers and health-care resources. BA has a recognizable latent or early symptomatic stage. There are well-established BA care pathways with clearly defined diagnostic features and acceptable treatment regimens. Timely intervention improves outcomes with proven cost-effectiveness. What is lacking is a single diagnostic screening laboratory test. Moreover, the incorporation of a BA screening test into current newborn dried blood spot cards has been hindered by the lack of an acceptable biomarker. It is not possible to measure direct or conjugated bilirubin from the dried blood spot cards that are currently used in newborn screening programs. The measurement of glycocholic acid, chenodeoxycholic acid, or other bile acids on blood spot cards have poor sensitivity and specificity and lack sufficient screening performance [34].

### 4.1. The UK “Yellow Alert” Educational Campaign

The first attempt towards early BA detection was the “Yellow Alert” educational program introduced in the UK in 1993 by the late Professor Alex Mowat at King’s College Hospital, in association with the UK Children’s Liver Disease Foundation and the Department of Health [35]. The aim of this national campaign was to ensure direct or conjugated bilirubin testing in all babies with persistent jaundice after two weeks of age. Despite worldwide efforts to raise awareness through educational programs directed towards health care professionals and the public, the laboratory investigation of newborns with prolonged jaundice has not been well-integrated into standard care practice for neonates.

### 4.2. BA Screening Using a Stool Color Card

In 1994, Matsui introduced a seven-colour panel stool colour card (SCC) (three abnormal stool colours) to the Maternal and Child Health Handbook that was distributed to all pregnant women in the Tochigi Prefecture in Japan (Figure 2, Matsui A. [36]). Before or at the routine 1-month newborn follow-up, mothers returned the completed SCC card to the attending physician, and all suspected cases were then referred for further examination. Between 1994 and 2011, a total of 313,230 newborns were screened and 34 cases of BA were diagnosed [37]. The card return rate was 84%. The SCC screening performed well (Table 1). The mean infant age at KPE was 60 days through the 19-year screening period, having decreased significantly from 70 days in the 1987–1992 historic cohort prior to screening. Of the eight patents who were missed by SCC screening at 1 month, two had been in the NICU, and abnormal stool colour was overlooked; three patients were recognized to have a pale stool colour, but their caregivers did not pursue further examination because the babies were not visibly jaundiced; one patient had not used the SCC; one patient had reportedly normal stool colour at the 1 month follow-up; and one patient was identified outside of the program.

Following a regional pilot study in 2002–2003, Taiwan was the first country to introduce a national universal screening program for BA in 2004 using the SCC [38]. Their first iteration of the SCC, a six-colour panel card with three abnormal stool colours, was later changed to a nine-panel card with six abnormal stool colours. The SCC is integrated into the Taiwan child health care booklet given to every neonate in the country. Mothers are asked to contact the screening centre by phone or fax when concerned about their infant’s stool colour. The SCC is checked by a physician at the routine 1 month health visit at the time of HBV vaccine delivery. During the Taiwan universal program of 2004–2005, there were a total of 422,273 births, and 75 BA cases were reported nationally [39]. In 2004 and 2005, 73% (29/40) and 97% (34/35) of the BA cases were successfully screened by the SCC before 60 days of age, respectively (Table 1). There were 187 false-positive cases reported with a transient pale stool colour. In 2004, 15 of the 40 BA cases had a KPE >60 days of age: eight patients had not used the SCC, one case had a delayed diagnosis by the health care professional, two suffered an erroneous judgment of stool colour, two had delayed identification of pale stool colour, and two had a delayed visit to the physician. In 2005, 9/35 cases had delayed surgery >60 days of age: three with delayed identification of pale coloured stool, three with delayed physician visits, one with incorrect judgment of stool colour, and one with a delay in diagnosis. In a five-year follow-up outcome study, the age >90-day KPE cohort had been virtually eliminated, and both the 3-month post-KPE jaundice-free rate and the 3-year jaundice-free native liver survival rate had improved from 35% to 68% and 32% to 57%, respectively, between the pre- and post-screening eras [40].

A recent SCC screening program in Sapporo, Japan in 2012 and in the Chao Yang district in Beijing, China in 2013 reported that both centers used a seven-colour panel card while having three abnormal coloured stools [46]. In Beijing, the SCC was distributed directly at maternity, and the family was advised to bring the stools and their infant to the SCC screening centre if abnormal stool colour was detected. Stool data were also verified directly with the family where the infant was aged 2 weeks, one month, and 1–4 months through a combination of mobile phone calls and text messaging, as well as the routine 42-day health check-up by pediatricians through the city’s neonatal screening system. In Sapporo, the SCC is integrated into the Maternal and Child Health Handbook that is distributed together with postcards and maternal health check-up tickets to all women during their pregnancy. If abnormal stools are detected, the families are instructed to bring the stools and their infant to the local hospital. For each infant, a completed postcard with the SCC data was collected at the one-month health checkup.

A large-scale prospective Canadian cohort study in the province of British Columbia (BC) found that distribution of the SCC at maternity was the most effective and highly cost-effective screening strategy [47]. The findings were confirmed at another care centre in Montreal, Quebec [48]. The BC provincial screening program was implemented in 2014. The SCC was initially a six-panel colour card with three abnormal stool colours identical to the Taiwan colour stool photos. Currently, a nine-colour panel card analogous to the Taiwan SCC is used. The SCC is given to families at the time of discharge from the maternity unit. In the case of home delivery, the SCC is given by the midwife to the family. Babies admitted to a NICU are excluded from the program and not issued a SCC. Parents are instructed to regularly monitor their infant stool colour at home using the SCC for the first 30 days after birth, and to contact the screening centre by phone or email with any concerns about their infant’s stool colour. All follow-ups are provided by a pediatric hepatologist. From 2014–2016, there were 87,583 births in British Columbia, and six cases of BA were identified [42]. The SCC screening successfully identified abnormal stool colour in 5/6 cases. In the one case of unsuccessful SCC screening, abnormal stool colour was not consistently recognized by the family and contact was not made with the screening centre. The family was instead seen by a physician because of prolonged jaundice, but no testing was done. The screening program instigated timely case referral to specialty care (defined as program screen success) in 3/6 BA cases. Of the three program screen failures, two families who correctly identified pale stools sought immediate consultation with their care providers; however, they were reassured and no further timely action was taken. In the third case, the infant was taken to the physician with complaints of jaundice but not of abnormal stool colour, and no investigations were performed. Most of these cases had late referral and delayed diagnosis with a median age of KPE of 116 days (49–184 days). The performance of the SCC screening was comparable to other national reports (Table 1). A new SCC is now utilized, having a highlighted statement to instruct physicians to order a fractionated bilirubin test for newborns who present with parental concerns about their infant’s stool colour.

National BA screening programs using an SCC have been introduced in Switzerland and Japan [36,49]. Small pilot studies with the SCC have been conducted in regions or municipalities in Brazil [50], Cairo, Egypt [51], Shenzen (China) [52], Northern Portugal [53], and Lower Saxony Germany [54]. However, to our knowledge, comprehensive feasibility and performance studies of the SCC in these locales have not yet been reported.

### 4.3. BA Screening Using a Stool Colour Smartphone App

Several centers have developed mobile smartphone applications designed to help parents and caregivers identify abnormal stool colour and prompt early referral to specialty care (Table 2). These applications have a touch-screen interface, utilize the smartphone camera, and apply specially designed colour analyzer software to assess the infant’s stool colour. Abnormal stool colour triggers a message to the user to seek consultation with their health care provider.

PoopMD^®^, developed at John Hopkins University in the United States and released in 2014, was the first stool colour application for iOS and Android devices [55]. The colour recognition software was based on the Taiwan stool colour card images converted to a 16-base colour pallet using RGB digital photo hexcodes. The interface targeted adolescent and young adult parents having an eighth-grade reading proficiency level. The accuracy of the mobile app was determined by seven expert pediatricians based on 34 photographs of infant stool. The application correctly identified all acholic stool photos without any false-negatives. While 11% of the photos were classified as indeterminate, none of the normal stools were identified as acholic (Table 2).

Baby Poop^®^ is an iOS-based application developed in Japan and released in 2016 [56]. A total of 54 BA and 100 non-BA stool images were collected to develop the colour detection algorithm. Both RGB and HSV (hue, saturation, and value) attributes were used in the stool colour analysis (Table 2). HSV was demonstrated to be an important component in the accurate identification of abnormal stool colour. The application, in Japanese only, had 100% sensitivity and specificity for the detection of BA based on a test sample of 40 stool pictures including five BA stools. A similar application is being developed in Shanghai [57].

Popòapp^®^, designed by an Italian team, is another mobile device application for both iOS and Android devices [43]. The colour analyzer algorithm is based on the Japanese seven-stool colour photo panel using an RGB digital colour system. After completing a baseline questionnaire, the user takes a picture of the stool. Results are categorized as “normal”, “abnormal”, or “uncertain”. Any stool colour that cannot be classified by the app is defined as “indeterminate”. The application was validated by four pediatric subspecialists using 160 stool samples from infants ≤6 months of age who had been admitted to an inpatient hepatobiliary service. The application performed well without any false-negative results (Table 2).

### 4.4. BA Screening Using Conjugated or Direct (Fractionated) Bilirubin

The first study to apply a fractionated bilirubin test for newborn BA screening was conducted in the UK in 1998 [44]. The investigators measured conjugated bilirubin levels in infants 4–28 days old using extra plasma collected from routine newborn screening. At the time, in Birmingham and other parts of the UK, routine newborn screening was based on liquid capillary blood specimens. In a follow-up prospective study of 23,214 patients using defined bilirubin cut-offs, testing had a sensitivity of 100%, specificity of 99.6%, and positive predictive value of 10.3% for the detection of BA [45]. Testing also identified other diseases, including Alagille Syndrome, alpha-1-antitrypsin deficiency, and panhypopituitarism. As mentioned previously, the dried “Guthrie” blood spots for newborn screening could not be used to test for fractionated bilirubin measurements.

In the United States, recent studies explored bilirubin testing in the first 24–48 h of life based on the observation that newborns with BA have elevated direct or conjugated bilirubin levels starting at birth [58]. The screening algorithm involves blood procurement for testing all infants before discharge from the newborn nursery. Infants with high levels of fractionated bilirubin were later tested as outpatients at the routine two-week well-child visit, and those with persistently high levels were referred for further evaluation. This screening algorithm for BA was tested in a pilot study of 11,636 infants and a larger follow-up study of 123,279 infants (Table 1, [59]). Screening resulted in significant improvements in the timing of the KPE. Screening also identified other diseases, including Alagille Syndrome, alpha-1-antitrypsin deficiency, progressive intrahepatic cholestasis, and choledochal cyst. Screening newborns for BA with fractionated bilirubin measurements is now also taking place in other US locations, including San Antonio, Salt Lake City and surrounding areas, and New Orleans [60].

## 5. BA Screening and Public Health Policy: The Challenge of Influencing Policymakers and Considerations for Program Implementation

Advocacy and the realization of a BA screening program requires a strong leadership team to champion the proposal and bring it to fruition. Meticulous and well-coordinated planning with local governmental authorities, newborn screening advisory committees (or similar groups if they exist), and other local experts in the field is an absolute necessity. It is recommended to have strong engagement among patients and their families to help garner support for the program by governmental representatives. Careful consideration of country-specific health care resources and capacities, particularly in the context of the program development plan is key. The choice of screening methodology that is most appropriate to the respective region, the required infrastructure for the operation of the program, and the optimal process for case follow-up are necessary first steps to consider before embarking on the implementation of the program.

There are now several publications to justify the need for a BA screening program that demonstrates the feasibility, efficacy, and cost-effectiveness of screening [27,42,47,61]. It is helpful to have pilot program study data from the respective region to inform stakeholders of the local BA outcome and infant age of KPE in the context of published reports from elsewhere in the world. Every country needs a “homegrown” approach to show that BA screening is needed, and a reliable, easy-to-apply, and suitable screening program for their own population can be proposed.

Despite the scientific evidence of the benefit of early diagnosis and management of BA patients, the creation of a SCC screening program is challenging. In Taiwan, the Department of Health Director became very supportive of the BA screening program after an enthusiastic explanation of the plan by Professor Mei Hwei Chang with the support of the community, care providers, and the local press (personal communication). Universal BA screening in Taiwan was approved following a pilot project that expanded from small regional sites to a national study. In Switzerland, the Swiss reference center for BA patients initiated a national feasibility screening program using the SCC in 2009. It took 10 years before SCC screening for BA was introduced into the Swiss national health booklet (www.paediatrieschweiz.ch, accessed on 24 December 2021) (personal communication). In Germany, the initiative for BA SCC screening was launched in 2016 by the Hannover group [54]. The physicians cooperated with a major German health insurance company and with the local Medical Association in their region in Lower Saxony to establish the screening project. This local initiative opened its doors at the German Federal Joint Committee, and the group was authorized to make binding regulations in the country. Negotiations are still underway for a SCC to be enclosed nationally in German children’s health booklets.

In Canada, health care is a federal charter, but newborn screening programs are provincially mandated and directed. A pilot study for the SCC program was first completed in British Columbia before it was supported by the provincial perinatal services and BC Ministry of Health. In France, a private parent organization, the Association Maladies Foie Enfants (AMFE), is promoting the screening of “yellow babies” using stool colour, and has launched many national information campaigns to encourage the screening mode called “*alerte jaune*” (“yellow alert”), including spots on national TV stations and annual national sensitization events (alertjaune.com, accessed on 29 December 2021). In the US, the choice of the diseases to be included in newborn screening involves coordination between federal and state policy makers [62]. At the federal level, the Secretary of the Department of Health and Human Services makes recommendations as to which diseases belong on the Recommended Uniform Screening Panel, or RUSP, based on expert opinions from the Advisory Committee on Heritable Disorders in Newborns and Children. However, individual states are not required to follow the RUSP. The state policy-makers are the ones who decide which diseases will be screened for in their state [2].

Each screening modality requires its own infrastructure and evidence-based data to support its adoption and successful implementation. BA screening using a SCC is simple and inexpensive, whereas the cost-effectiveness of fractionated bilirubin testing is still under investigation in the US. In developing countries, screening using an SCC may be a preferred methodology because of its low cost. For smartphone applications to be effective for BA screening in any given region, comprehensive and widespread distribution and the use of mobile phone technology is necessary, and the quality of the screen needs to meet the standards. Screening applications require further validation studies before they should be adopted. In the US, a SCC screening program may be more difficult to implement universally, given the decentralized structure of the health care system.

For SSC-based screening programs, Taiwanese, Japanese, Canadian and Swiss program leaders have emphasized the importance of having comprehensive educational seminars about the screening program before the program launch. These should be directed towards maternity nurses, other health care professionals on the maternity wards, midwives, and community newborn care providers. The use of webinars, virtual teaching sessions, or scripted educational sheets to uniformly explain the screening process to families is an essential requirement for a program’s success. Considerations must be given to families’ first language and socioeconomic status to ensure the instructions are clearly understood. Recent input from developing countries support the concept that this screening process is simple and easily understood even in lower socioeconomic groups and those with illiteracy [51].

The SCC requires validated stool colour photos. The Japanese have emphasized the need for reproducible digital photographic images with CYMK-based metafiles to ensure the quality of the stool colours and reproducibility of card-printing [36]. The colour and hue of stool photos on a computer or a smartphone will depend on the resolution of the colour screen, while printouts of the SCC depend on the quality of the colour printer and paper. Caution is advised against using home printers for card distribution. Instead, professionally printed cards on quality paper or stool photos incorporated into a professionally printed health booklet is recommended. In the case of smartphones, the use of applications having valid and real-life proven software colour analytics is necessary.

There are other nuances with the current SCC screening programs. Japan and Taiwan have differing screening processes. The former includes check points at infant age 2 weeks, one month and 1–4 months, while the latter only includes a 30-day follow-up. In Japan, the screening program is under the jurisdiction of each local government and the screening program policies differ regionally. Additionally, the SCC is contained within the Maternal and Child Health handbook, which is distributed to pregnant women in the antenatal period during pregnancy and prior to delivery. In British Columbia and Switzerland, the SCC is distributed at the time of discharge from maternity. Families are instructed at that time to monitor the stool for the first 30 days of life. In British Columbia as in Switzerland there is no formal check-point time with the screening centre or other health care professional. In the US, given the ease of fractionated bilirubin testing newborns in the first 24–48 h of life before hospital discharge, implementation of this BA screening program is now being considered in various US centers.

## 6. Limitations of the Current BA Screening Programs

Each of the existing SCC screening programs have been hampered by delayed follow-up of patients screening positive for stool colour. This is a major concern and efforts need to be made to ensure prompt recognition and referral of these patients. There are no universally accepted BA diagnostic algorithms, and several of the potential laboratory or imaging studies to diagnose BA take a long time to perform and interpret. Cases identified through a BA screening program need the corresponding infrastructure to facilitate prompt referral to a centre with expertise in BA assessment and management instead of leaving the workup to community care providers.

Two important challenges to the implementation of fractionated bilirubin screening are variations in normal values for fractionated bilirubin assays between laboratories and the process for communicating abnormal results in a timely manner [60]. Assay variations arise because hospital laboratories use either direct or conjugated bilirubin methodology. While conjugated bilirubin methodology is consistent across sites, direct bilirubin methodology can differ slightly from site to site, depending on the reaction conditions used. As a result, providers must interpret direct bilirubin levels using site-specific reference intervals rather than a universal standard that can be used at all sites. Communication challenges arise because the screening algorithm requires outpatient providers to know when an abnormal result occurs in the newborn nursery. In the US, medical care is decentralized and there is no universally shared medical record. As a result, primary care providers may not have access to complete information about the initial newborn hospitalization, including the results of the first direct or conjugated bilirubin measurement. The outpatient providers may not be aware of which infants tested positive and who requires timely follow-up.

Finally, as in other screening programs, there is a risk that BA screening will unnecessarily increase parental anxiety with false-positive cases [63]. Strategies to reduce anxiety focus on improving ways to communicate information to parents. A recent study used a parental questionnaire to investigate parental anxiety with the SCC [49]. The study showed that of the respondents (*n* = 109), most did not experience negative feelings when using the SCC or discussing liver diseases with their physician in the context of SCC use.

## 7. Conclusions

BA affects infants around the world, with birth prevalence showing regional and, in some cases, seasonal variation. Despite its unknown etiology, the importance of early treatment with the KPE in delaying or avoiding liver transplant is well-established. Newborn screening of BA represents a powerful way to ensure prompt diagnosis of affected infants. Multiple screening modalities have been examined, with the SCC and direct or conjugated bilirubin screening being implemented on the largest scales and also being the most promising. The SCC has been incorporated into national screening strategies and adapted to smartphone apps, whereas fractionated bilirubin screening is still under investigation. Future studies are needed to determine how best to implement screening strategies more widely, taking into account region-specific resources and variations in the process of policy decisions that are needed to ensure broader adoption. Future studies to establish the diagnostic algorithms for BA are required to ensure that screen-positive infants are evaluated efficiently and receive the KPE in a timely manner, ideally before 30 days of life.

## Figures and Tables

**Figure 1 jcm-11-00999-f001:**
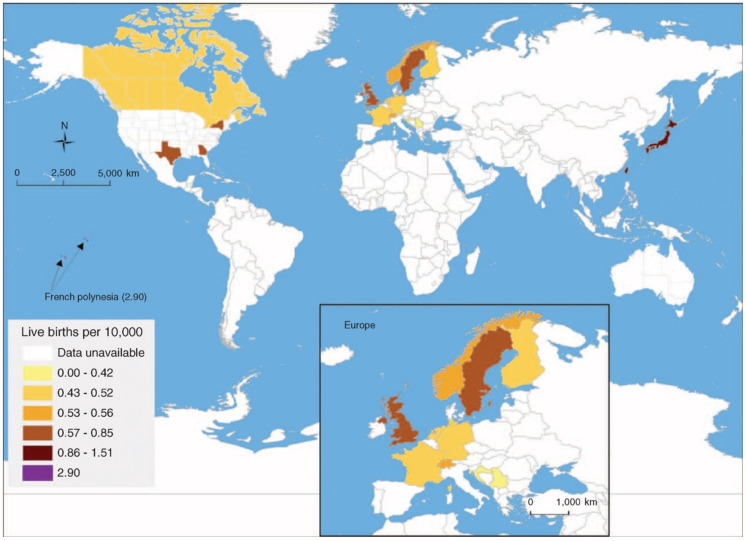
Worldwide incidences of biliary atresia (Jimenez–Rivera et al. [3], © 2013 reprinted by the permission of Wolters Kluwer Health).

**Figure 2 jcm-11-00999-f002:**
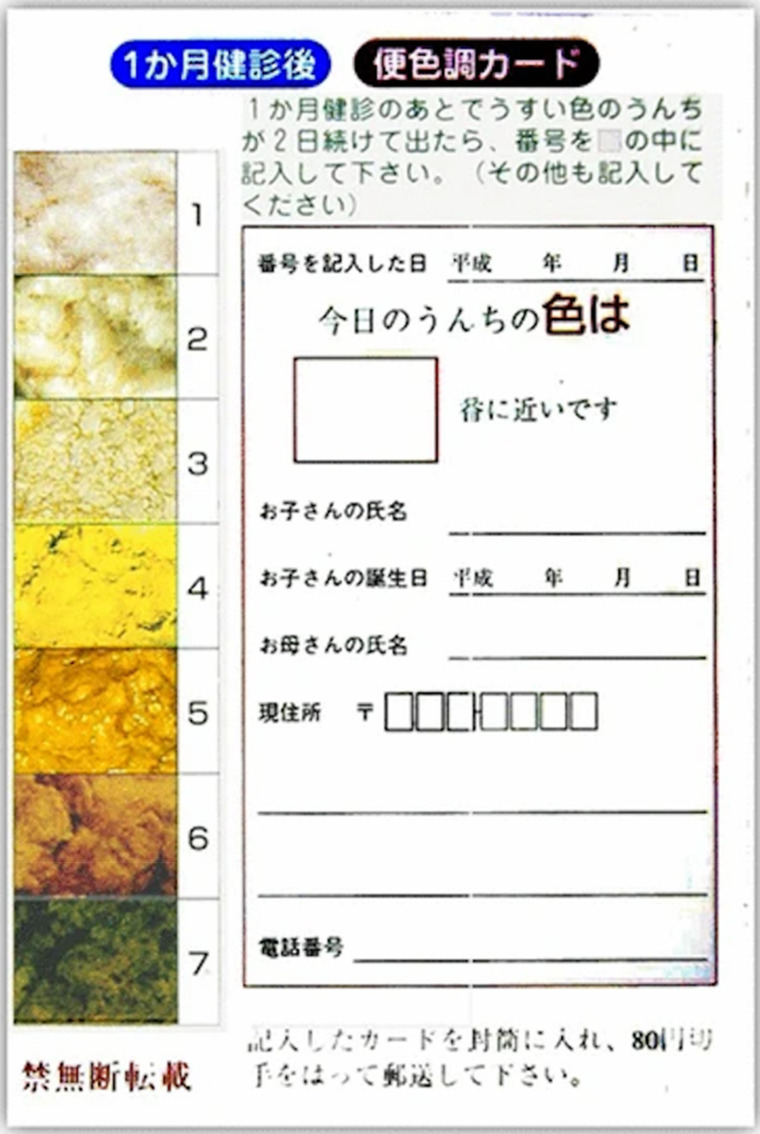
The Matsui stool colour card. This is the first version of SCC in Tochigi Prefecture, Japan. It was delivered to all pregnant women together with a “Maternal and Child Health Handbook”. Stool colors were numbered. Images 1–3 were pale-pigmented, and images 4–7 were bile-pigmented stools. A mother is asked to compare the colour of her infant’s stool to that of the card, to fill in a corresponding number just before the 1-month health checkup, and to hand it to the attending doctor. When pale-pigmented stools were suspected, the doctor reported to the SCC office by telephone or fax immediately; otherwise SCC were returned to the office by post weekly. The figure is provided to respectfully acknowledge the ground-breaking work of Professor Akira Matsui who passed away in 2020. There are now several versions of the SCC in many languages that are used for BA screening around the world. (Matsui [36], © 2013 reprinted by the permission of Springer Nature).

**Table 1 jcm-11-00999-t001:** BA screening performance.

Stool Colour Card Screening								
Country	Year	# Screened Patients	BA Cases	Sensitivity (%)	Specificity (%)	PPV (%)	NPV (%)	KPE Age Pre-/Post Screening
Taiwan * [39] Universal national program	2004–2005	422,273	75	84	99.9	22.5	99.9	<60 days: 47%/67% 0>91 days post screening
Japan ^#^ [37] TochigiPrefecture	1994–2011	313,230	34	76.5 (62.2–90.7)	99.9 (99.9–100.0)	12.7 (8.2–7.3)	99.9 (99.9–99.9)	67/56 (median days) 25%/11%>80 days
Chaoyang District Beijing ^†^ [41]	2013–2014	29,799	4	50	99.9	4.5	99.9	n/a
Canada ^§^ [42] British Columbia	2014–2016	87,583	6	83	99.9	6	99.9	n/a
* Diagnostic accuracy statistics for detecting BA by 60 days of life; ^#^ Diagnostic accuracy statistics for detecting BA by 1 month of life; ^†^ Diagnostic accuracy statistics for detecting BA by 4 months of life; ^§^ Diagnostic accuracy statistics for detecting BA by 1 month of life; n/a = not available.								
**Fractionated Bilirubin Screening**								
Country	**Year**	**# Screened Patients**	**BA Cases**	**Sensitivity**	**Specificity**	**PPV**	**NPV**	**KPE Age Pre-/Post Screening**
UK * [43]	1995–1997	23,214		100.0 (76–100)	99.5 (99.5–99.6)	10.3 (5–16)	n/a	n/a
US ^#^ [44]	2013–2014	11,636	2	100.0 (20–100)	99.9 (99.8–99.9)	18 (3–52)	n/a	n/a
US ^#^ [45]	2015–2018	123,279	7	100.0 (56–100)	99.9 (99.9–99.9)	5.9 (3–12)	100 (100–100)	56/36
* Diagnostic accuracy statistics for detecting BA by 28 days of life (last follow-up test for BA patients performed on day of life 22); ^#^ Diagnostic accuracy statistics for detecting BA by 2 weeks of life in a two-stage screening approach (first test in newborn period, second test at 2 weeks of life if first test abnormal); n/a = not available.								

**Table 2 jcm-11-00999-t002:** Mobile device screening application (Angelico et al. [43], © 2017 reprinted by the permission of Sage Publications).

Characteristics	PoopMD	Baby Poop	PopòApp
Year	2015	2017	2020
Country	USA	Japan	Italy
Reference	[14]	[16]	Current study
Programming language	Java		Java
Operating system	iOS/Android	iOS	iOS/Android
Source of pictures	Previously validated and recorded	Pre-existing images	Newly acquired images taken with the Pop6App
Establishment of the gold standard for stool color	ISCC	Pre-existing BA and non-BA stool images	ISCC
Color analyzer system	RGB parameters	RGB and HSV parameters + ma chine learning process	RGB system + machine learning process
Clinical assessment of the App	Agreement between 6 doc-tors who revisited the pictures	Performance tested with pre-classi-fied images	Real-time assessment by 4 doctors who took the images (agreement between 4 doctors)
Classification of stool color	Acholic, cholic, indeterminate	Acholic, cholic	Acholic, cholic, uncertain, indeterminate
Number of pictures for Accuracy test of the App	34	40	160
– Acholic	7	5	60
– Normal	24	35	63
– Uncertain			16
– Indeterminate	3		21
Sensitivity (95% CI)	100%	100% (48–100%)	100% (93.9–100.0%)
Specificity (95% CI)	89%	100% (90–100%)	99% (94.6–99.9%)

BA: biliary atresia; CI: confidence intervals; HSV: hue-saturation-value; ISCC: infant stool color card; RGB: red-green-blue.

## Data Availability

Not applicable.

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
