# Peer review of "Biliary Atresia in 2021: Epidemiology, Screening and Public Policy"

_jcm, 2022, doi:10.3390/jcm11040999_

Round 1
Reviewer 1 Report
Thank you for the opportunity to review the manuscript entitled, "Biliary Atresia in 2021: Epidemiology, Screening and Public 1 Policy”. Authors focused on biliary atresia from a global public health perspective including disease epidemiology, current national screening programs and their impact on outcome, as well as new and novel BA screening initiatives. However, there is not a novelty. Therefore, the manuscript may be able to benefit from major revision if the manuscript is revised.
Major revision
- There is not a novelty because of many previous reports. We should show disease epidemiology, current national screening programs and their impact on outcome from non-North American, European, and Asian countries.
- It has been reported that various metabolites or proteins, including serum hyaluronic acid, apolipoprotein C3, interleukin-6, interleukin-8, matrix metalloproteinase-7, urine sulfate conjugated bile acid, urine oxysterols, and stool secondary bile acids have identified potential diagnostic or screening markers for biliary atresia. You should discuss about these markers as the possibility of screening.
Author Response
Reviewer 1:
Thank you for the opportunity to review the manuscript entitled, "Biliary Atresia in 2021: Epidemiology, Screening and Public 1 Policy”. Authors focused on biliary atresia from a global public health perspective including disease epidemiology, current national screening programs and their impact on outcome, as well as new and novel BA screening initiatives. However, there is not a novelty. Therefore, the manuscript may be able to benefit from major revision if the manuscript is revised.
Major revision: There is not a novelty because of many previous reports. We should show disease epidemiology, current national screening programs and their impact on outcome from non-North American, European, and Asian countries. It has been reported that various metabolites or proteins, including serum hyaluronic acid, apolipoprotein C3, interleukin-6, interleukin-8, matrix metalloproteinase-7, urine sulfate conjugated bile acid, urine oxysterols, and stool secondary bile acids have identified potential diagnostic or screening markers for biliary atresia. You should discuss about these markers as the possibility of screening.
We thank Reviewer 1 for the comments, however we disagree with them. This manuscript is an invited review and it is not aiming to be a novelty new research paper. It focuses on current, clinically applicable and implemented screening strategies and not on potential new ones. This paper is, to our knowledge, the first to describe the current state of biliary atresia (BA) screening using recognized, clinically applicable and nation-wide implemented “markers”, such as conjugated or direct (fractionated) bilirubin, the stool color card, or web applications.
We believe that it is beyond the scope of our paper to include and discuss the BA-markers mentioned by Reviewer 1. Again, this review focuses on existing, implemented methods, not on theoretical methods that may pan out in the future. Most of what Reviewer 1 suggests for BA screening are actually putative diagnostic or prognostic markers for BA. To the best of our knowledge none of them have been studied or reported as a BA screening tool in large populations or national programs.
To make the purpose of our paper clearer to the reader, we added the following at the end of the Introduction; The aims of this article are to provide up- to-date knowledge of the epidemiology of biliary atresia, to focus on the current clinically applicable BA screening strategies implemented in large populations or national programs and to describe the challenges of BA screening in the context of public health policy.
This said, the matters raised by Reviewer 1 may be discussed elsewhere in the upcoming special journal on BA, perhaps in another chapter on BA diagnostics?
Reviewer 2 Report
This is a very well written and nice paper to read. The authors are well known on this field. They describe an overview of the available data concerning screening for BA, a devastating disease if not treated early.
The only correction I suggest is in line 38, page 1. the authors stated "a loop of small bowel is brought to the porta". it is better to write "porta hepatis", which is the correct name of the place where the porto-entero anastomosis is performed.
Author Response
Reviewer 2:
- This is a very well written and nice paper to read. The authors are well known on this field. They describe an overview of the available data concerning screening for BA, a devastating disease if not treated early.
We thank Reviewer 2 for this comment.
- The only correction I suggest is in line 38, page 1. the authors stated "a loop of small bowel is brought to the porta". it is better to write "porta hepatis", which is the correct name of the place where the porto-entero anastomosis is performed.
We thank Reviewer 2 for this suggestion and adapted the manuscript accordingly.
Round 2
Reviewer 1 Report
I have no comments.